# Examining the Association Between Food Insecurity, Food Literacy, and Food Intake Among Low-Income Adults in Jeddah, Saudi Arabia: A Cross-Sectional Study

**DOI:** 10.3390/foods14173078

**Published:** 2025-09-01

**Authors:** Mahitab Hanbazaza, Lama Alaklabi

**Affiliations:** Department of Food and Nutrition, Faculty of Human Sciences and Design, King Abdulaziz University, Jeddah 22258, Saudi Arabia; lalbeeshi@stu.kau.edu.sa

**Keywords:** food security, food literacy, food intake, Saudi Arabia, adult

## Abstract

This cross-sectional study assessed the prevalence of food insecurity and food literacy and examined food intake among low-income Saudi families. It provides valuable insights into the association between food literacy, food insecurity, and food intake, contributing to a better understanding of the food-related challenges faced by vulnerable families. Conducted in June 2024, this study involved 483 low-income adults in Jeddah, Saudi Arabia. Data were collected through an online questionnaire distributed via charitable organizations. Overall, 35% of the participants experienced severe food insecurity, while 91.5% demonstrated adequate food literacy. No significant association was found between food literacy and food insecurity levels (*p* = 0.586). However, severe levels of food insecurity were significantly associated with reduced intake of fruits, vegetables, dairy products, meat, bread and cereals, sugar-sweetened beverages, potato crisps or salty snacks, sweets, and fast food (*p* < 0.001). Additionally, a significant difference was observed in the consumption of potato crisps or salty snacks, with individuals exhibiting adequate food literacy consuming slightly more than those with poor food literacy (0.25 ± 0.28 vs. 0.21 ± 0.30, *p* = 0.04). These findings highlight the need for interventions that not only build food knowledge but also address affordability, access, and cultural food norms among low-income Saudi families.

## 1. Introduction

In recent years, the number of low-income households in Saudi Arabia, defined as those earning less than SAR 5000 (approximately USD 1333) per month, has increased due to rising living expenses and economic pressures [1]. Low socioeconomic status is commonly linked to poorer dietary quality, a higher prevalence of obesity, and more sedentary lifestyles, all of which contribute to elevated risks of chronic diseases worldwide [2,3]. Globally, individuals with lower incomes tend to consume lower-quality diets, characterized by reduced fruit and vegetable intake and greater reliance on energy-dense, nutrient-poor foods such as sugar-sweetened beverages [4]. These dietary patterns exacerbate the economic burden of food insecurity, which occurs “when individuals or households lack sufficient socioeconomic or physical access to adequate, nutritious, and safe food that meets their dietary needs and preferences for a healthy, active life” [5]. Food insecurity is a major public health concern due to its well-documented links to poor physical and mental health, placing a significant strain on healthcare systems [5].

Food insecurity remains a persistent problem worldwide. In South Africa, 15% of the households reported inadequate food access in 2021, while 6% reported severely inadequate access [6]. In 2023, 28.2% of the population in Latin America and the Caribbean experienced moderate or severe food insecurity [7]. In the Arab world, nearly one-third of the Arab population faced food insecurity in 2020, according to the Food and Agriculture Organization (FAO) [5]. In Saudi Arabia, studies among vulnerable groups indicate high prevalence rates. For instance, a study conducted in Jeddah reported that 50% of low-income families experienced some level of food insecurity [8].

Given the complex nature of food insecurity, attention has shifted toward the potential role of food literacy. Food literacy encompasses the skills, behaviors, and knowledge needed to organize, manage, choose, prepare, and consume food in ways that meet dietary needs and support informed choices. It includes the ability to read food labels, interpret nutritional information, follow food safety practices, and prepare healthy meals in line with dietary guidelines [9]. Improving food literacy may alleviate certain aspects of food insecurity by enabling individuals to maximize the value of their available income [10]. However, its impact is limited, as education alone cannot reduce food costs or solve the broader economic issues underlying food insecurity [11].

The cultural and religious context in Saudi Arabia, including festivals such as Eid al-Fitr and Eid al-Adha, Ramadan, and the Hajj season, strongly influences dietary behaviors and food security dynamics. For example, Ramadan fasting leads to significant changes in meal timing and food choices, with traditional eating patterns shifting toward richer, more elaborate meals [12]. Additionally, the Islamic practice of Zakat, or almsgiving, plays a critical role in supporting food security among low-income households by influencing both food distribution and community interactions [13]. These cultural and religious practices provide an important lens through which to understand how food security and food literacy function in the Kingdom and shape individual dietary behaviors.

Previous research suggests that food insecurity may be reinforced by poor food literacy skills [14]. Enhancing food literacy is believed to improve food preparation knowledge and resource management, with the potential to reduce some aspects of food insecurity [14]. Despite this recognized importance, few studies have examined the association between food insecurity, food literacy, and food intake among low-income families in Saudi Arabia. Addressing this gap is essential for developing effective interventions and policies that promote equitable access to nutritious food. Therefore, the purpose of this study was to assess the prevalence of food insecurity and food literacy and to examine food intake among low-income Saudi families.

## 2. Materials and Methods

### 2.1. Design and Participants

This cross-sectional study was conducted in June 2024 and included 483 low-income adults residing in Jeddah, Saudi Arabia. Jeddah, one of the country’s largest and most socioeconomically diverse cities, has a unique demographic and cultural composition, making it an ideal setting to investigate the interplay between food insecurity, food literacy, and dietary intake among low-income families.

The minimum required sample size was 384, calculated using an online sample size calculator with a 95% confidence level, 90% power, and the assumption that 50% of the population was exposed to the risk factor, aiming to detect an odds ratio of 2 [15]. The formula for the calculation is as follows:$$\mathrm{Minimum}\;\mathrm{sample}\;\mathrm{size},\;n\;=\;\frac{z^{2}\times p\times (1-p)}{d^{2}}=\;\frac{{\left(1.96\right)}^{2}\times 0.5\times (1-0.5)}{{\left(0.05\right)}^{2}}=384.16\approx 384$$

The participants included Saudi adults aged 18 years and older who were beneficiaries of charitable organizations. No exclusion criteria were applied. As a token of appreciation, those who completed the survey received USD 14 purchase vouchers. This study followed ethical guidelines, including obtaining informed consent, ensuring anonymity and confidentiality, and receiving approval from the Unit of Biomedical Ethics Research Committee at King Abdulaziz University, Jeddah (Reference No. 131-24).

### 2.2. Data Collection

Data were collected via an online Arabic questionnaire distributed through charitable organizations closely engaged with low-income communities. Face validity was assessed with 10 individuals to ensure clarity and consistency, while 5 nutrition experts evaluated content validity.

The questionnaire gathered socio-demographic information, including age, gender, marital status, education level, occupation, income, household size, number of children, housing structure, housing conditions, food preparation practices, and receipt of charity support.

#### 2.2.1. Food Security Assessment

Household food security was assessed using the Arabic version of the Food Insecurity Experience Scale (FIES) [16]. This tool evaluates food insecurity severity through eight self-reported questions about conditions and behaviors experienced in the previous 12 months. Responses were coded as Yes = 1, and No = 0; “don’t know” or “refused” responses were considered missing data and were excluded. Scores ranged from 0 to 8 and were categorized as food secure (0), mild food insecurity (1–3), moderate food insecurity (4–6), or severe food insecurity (7–8).

#### 2.2.2. Food Literacy Assessment

Perceived food literacy was measured using the Arabic version of the Short Food Literacy Questionnaire (SFLQ) [17], adapted from the version previously pilot-tested by Bookari (2023) to align with Saudi dietary guidelines [18]. The instrument includes 12 self-rated items that assess functional, interactive, and critical skills on a four- or five-point Likert scale. Scores ranged from 7 to 52, with higher scores indicating higher levels of food literacy.

#### 2.2.3. Food Intake Assessment

Food intake was assessed using two instruments. The first assessed consumption of various food groups [19], while the second assessed intake of sugar-sweetened beverages/soft drinks, salty snack foods, sweets, and fast foods [20]. Response options included “I don’t eat or drink it,” “less than once a week,” “once a week,” “2–3 times a week,” “4–6 times a week,” and “daily.” For analysis, categorical frequency responses were converted into continuous data by assigning approximate values in times per day (0 = never, 0.07 = less than once a week, 0.14 = once a week, 0.36 = 2–3 times a week, 0.71 = 4–6 times a week, and 1 = daily).

### 2.3. Data Analysis

Data analysis was conducted using IBM SPSS Statistics version 26.0 for Windows. Continuous variables were presented as means ± standard deviations (SDs), while categorical variables were reported as frequencies and percentages. For variables with non-normal distributions, the Mann–Whitney U and Kruskal–Wallis tests were used. For comparisons among more than two groups, the Kruskal–Wallis test was followed by pairwise post hoc tests with Bonferroni correction when significant differences were observed. Categorical variables were analyzed using the chi-square test with linear-by-linear association or Fisher’s exact test, as appropriate. All tests were two-tailed, with statistical significance set at *p* < 0.05.

## 3. Results

### 3.1. Participants’ Socio-Demographic Characteristics

Table 1 presents the demographic characteristics of the participants. Of the 483 individuals who participated in this study, the majority were female (89.2%) and aged 35–44 years (36.9%). In terms of education, about 32.9% had completed high school, while only 1.0% held postgraduate degrees. Marital status revealed that nearly half of the participants were widowed (48.2%). Most were unemployed (88.6%), with only 7.2% employed full-time. Regarding family structure, over half (57.1%) lived in households with five or more people, and 52.8% reported a monthly family income of less than SAR 3000. In terms of living situation, most were renters (73.9%). Regarding meal preparation, 35.4% reported preparing all their meals.

### 3.2. The Prevalence of Food Insecurity and Food Literacy

The prevalence of mild, moderate, and severe food insecurity was 17.9%, 25.3%, and 35.7%, respectively, while 21.1% of the participants were food secure (Figure 1).

In terms of food literacy, 8.5% of the participants had poor food literacy, and 91.5% had adequate food literacy (Figure 2).

### 3.3. Association Between Demographics and Food Literacy Levels

Appendix A describes the association between food literacy levels and various demographic and socioeconomic factors. A significant gender difference was observed, with 93.3% of the females demonstrating adequate food literacy compared to 76.9% of the males (*p* < 0.001). Age also had a notable effect: 31.6% of the participants aged 18–24 were classified as having poor food literacy, compared to lower percentages in the older age groups (*p* < 0.001). Marital status was another significant factor, as 32.3% of the single individuals had poor food literacy compared to only 8.9% of the married participants (*p* < 0.001). The participants who prepared no meals had a particularly high rate of poor food literacy (37.8%) compared to those who prepared all their meals (8.8%), a difference that was statistically significant (*p* < 0.001). No significant associations were found with education level, employment status, number of children, household size, housing structure, family income, or living situation.

### 3.4. Association Between Demographics and Food Insecurity Levels

Appendix A outlines the association between food insecurity levels and various demographic and socioeconomic factors. Significant differences in food insecurity were related to age (*p* = 0.014), education level (*p* = 0.014), marital status (*p* = 0.012), number of children (*p* = 0.031), and family income (*p* < 0.001). The younger adults aged 18–24 were most affected, with 52.6% experiencing severe food insecurity. The participants with primary education showed the highest prevalence of severe food insecurity (39.2%). In terms of marital status, the single individuals were significantly more likely to be food secure (35.5%), while the married and divorced participants were more likely to be severely food insecure (38% and 43.2%, respectively). Additionally, a significant association was found between food insecurity and the number of children, with the families having more than five children experiencing the highest rate of severe food insecurity (44.3%). For family income, those earning less than SAR 3000 per month (44.4%) were most affected by severe food insecurity. Meal preparation frequency was also significant; the participants who prepared no meals or some meals had higher rates of severe food insecurity (45.9% and 46.4%, respectively).

### 3.5. Association Between Food Literacy and Food Insecurity Levels

Table 2 describes the association between food literacy and food insecurity levels. Among the individuals with poor food literacy, 15% were food secure, 25% had mild food insecurity, 20% had moderate food insecurity, and 40% experienced severe food insecurity. For the individuals with adequate food literacy, 21.7% were food secure, 17.3% had mild food insecurity, 25.8% had moderate food insecurity, and 35.3% experienced severe food insecurity. There was no statistically significant association between food literacy and food insecurity levels (*p* = 0.586).

### 3.6. Comparison of Food Consumption Between Individuals with Poor and Adequate Food Literacy

A comparison of food consumption between the individuals with poor and adequate food literacy revealed no significant differences in the intake of most food items, as shown in Table 3. The consumption of fruits, vegetables, dairy products, meat, legumes, bread and cereals, sugar-sweetened beverages, sweets, and fast food did not differ significantly between the two groups (*p* > 0.05). However, a significant difference was found in the consumption of potato crisps or salty snacks, where the individuals with adequate food literacy consumed these slightly more than those with poor food literacy (0.25 ± 0.28 vs. 0.21 ± 0.30, *p* = 0.04).

### 3.7. Comparison of Food Consumption Between Different Levels of Food Insecurity

Significant variations in food consumption were observed across the different levels of food insecurity (Table 4). The individuals experiencing severe food insecurity consumed significantly fewer fruits, vegetables, dairy products, meat, bread and cereals, sugar-sweetened beverages, potato crisps or salty snacks, sweets, and fast food compared to the food-secure individuals (*p* < 0.001 for most comparisons). As food insecurity increased, the consumption of these food items gradually decreased. The intake of legumes also varied significantly (*p* = 0.014). These results indicate that more severe levels of food insecurity are associated with reduced intake of a wide range of food items.

## 4. Discussion

This study found that most participants had adequate food literacy, while a smaller proportion demonstrated poor food literacy. Females exhibited higher food literacy than males. Over one-third of the participants experienced severe food insecurity. Younger individuals, particularly those aged 18–24, were more likely to have poor food literacy and experience severe food insecurity. No significant association was observed between food literacy and food insecurity levels. However, higher rates of poor food literacy and severe food insecurity were associated with a lower frequency of meal preparation. Regarding food consumption, significant differences were limited to the intake of potato crisps or salty snacks, with those exhibiting adequate food literacy consuming these items more frequently. Additionally, more severe food insecurity correlated with reduced intake of various foods, including fruits, vegetables, dairy products, and meat.

Examining participant demographics, the majority were female, aged 35–44 years, and had completed high school. Comparable findings were reported in a study aiming to improve food literacy among low-income adults, in which most participants were female and aged 36–45, though a larger proportion in that study held diploma degrees [21]. Another study focusing on healthy eating and cooking skills of food-insecure adults also found that the majority were female, with a mean age of 42 years, and nearly half possessed post-secondary education [22]. The similarity in demographic characteristics between the present and previous studies underscores the importance of gender, age, and educational level in shaping outcomes related to food insecurity and food literacy interventions.

In terms of food security status, most participants in this study experienced moderate to severe food insecurity, with only 21.1% classified as food secure. Similar levels of food insecurity have been documented in other studies involving low-income adults. For example, research by Hanbazaza and Mumena (2022) among Saudi low-income women found that a substantial percentage experienced mild to severe food insecurity [8]. Likewise, West et al. (2020) reported that 61.9% of low-socioeconomic Australian adults experienced low food insecurity at baseline, with 38.1% being food secure [20]. These findings highlight the vulnerability of low-income populations to food insecurity due to factors such as insufficient income, unemployment, and limited resources.

Contrastively, most participants in the current study demonstrated adequate food literacy, with only 8.5% having poor food literacy. These results differ from those of Bookari (2023), where 46% of Saudi parents had poor food literacy, especially among low-income parents, who were 40% more likely to be food illiterate [18]. Furthermore, a study assessing food literacy among female-headed households in a disadvantaged community in western Honduras indicated a moderately high overall level of food literacy [23]. Several factors might explain these discrepancies, including methodological differences and variations in sample characteristics. The high proportion of adequate food literacy in the present study may reflect participants’ involvement with charitable organizations that offer educational campaigns to raise awareness.

Food literacy is a multifaceted concept influenced by demographic factors, such as income, gender, age, and educational level. Murakami et al. (2022) found that females possessed higher food literacy than males, aligning with the current findings [24]. In this study, poor food literacy was more prevalent among participants aged 18–24 and less common in older age groups, contrary to Murakami et al. (2022), who reported greater food literacy among younger adults [24]. These differences suggest that specific factors in Saudi Arabia may influence this relationship. Younger low-income adults may have limited cooking experience, depend more on affordable, ready-made foods, or reside in households where older family members make most food-related decisions—limiting opportunities to develop food literacy skills. Moreover, many in this age group have unstable incomes or are still completing their education, heightening their risk of food insecurity. Cultural norms around food preparation and household roles may further contribute to these differences.

No significant associations were identified between food literacy and either educational level or family income in the present study. Bookari (2023) reported contrasting results, indicating that university-educated participants were more likely to be food illiterate than those with lower education levels. Additionally, those with no income or earning less than SAR 3000 per month showed poorer food literacy [18]. A cross-sectional study on Iranian adults aiming to evaluate the social determinants influencing food literacy found that women scored higher in food literacy than men, with significant predictors including gender, age, education, marital status, social media use, and attitudes toward social media accounting for 16.2% of the variance in food literacy [25]. Differences in these findings could stem from social and educational factors that significantly affect food literacy.

The present study found that food insecurity was significantly associated with educational level, marital status, number of children, and family income. Althumiri et al. (2021) similarly reported that low-income participants were more likely to experience severe food insecurity, and households with five or more children had a significantly higher risk compared to those without children [26]. In contrast, Hanbazaza and Mumena (2022) found no significant association between food insecurity and socio-demographic characteristics, which contradicts the present findings [8]. Variations in methodology and demographic composition may account for these discrepancies.

Although recent research has examined the link between food insecurity and food literacy, this study found no significant association between them. In contrast, Begley et al. (2019) reported that food literacy-related behaviors, such as planning, management, shopping, preparation, and cooking, were statistically associated with food insecurity [14]. Similarly, Jomaa et al. (2022) found that food-insecure participants were less likely to engage in food literacy practices, like selecting, purchasing, and preparing healthy foods and nutritionally balanced meals [27]. Notably, both the present study and Begley et al. (2019) found that individuals aged 18–24 tend to experience poor food literacy, which contributes to severe food insecurity [14]. Additionally, this study indicated that high rates of poor food literacy and severe food insecurity correlate with less frequent meal preparation. However, a cross-sectional study of young Canadian adults revealed that even with adequate food literacy and cooking skills, food insecurity can still impede the ability to buy healthy ingredients and prepare meals regularly [28]. The sample in the present study differs from those in previous studies with respect to cultural and social factors—elements that play a significant role in how individuals interact with food; for instance, women in Saudi Arabia may possess high food literacy but face challenges in improving food security due to economic dependence on men, though the reverse may also occur [18]. This interplay between knowledge and cultural factors may help explain the lack of association between food literacy and food insecurity observed in this study. Methodological differences in study populations may also contribute to these divergent findings.

Regarding food consumption, the current study found no significant differences in the intake of most food items between individuals with poor and adequate food literacy, except for potato crisps or salty snacks, which were consumed slightly more by those with adequate food literacy. This outcome could be explained by the possibility that individuals with adequate food literacy who have busier lifestyles may opt for snacks such as potato crisps or salty snacks due to time constraints. Additionally, unhealthy snacks are often cheaper and more widely available than healthier alternatives. Those with adequate food literacy may also possess better nutritional knowledge but still consume unhealthy snacks as a treat [29]. However, these findings contrast with Yoo et al. (2023), who reported higher intake of fruits, vegetables, and whole grains among food-literate individuals [30]. Similarly, Lee et al. (2023) found that individuals with high food literacy were less likely to have insufficient intake across all food groups, which diverges from the present results [31]. This difference may be attributed to this study’s focus on low-income families.

Examining the association between food security and food consumption, this study found that individuals experiencing severe food insecurity consumed significantly fewer fruits, vegetables, dairy products, meat, bread and cereals, sugar-sweetened beverages, potato crisps or salty snacks, sweets, and fast food compared with food-secure individuals. These findings partially align with Araújo et al. (2018), who reported that food insecurity negatively affects fruit and vegetable consumption [32]. In contrast, Begley et al. (2019) found that food-insecure individuals had a higher frequency of fast food and soft drink consumption and a lower intake of fruit, with no significant differences in vegetable intake [14]. Differences in demographics and geographic characteristics between study samples may account for these variations, as such factors strongly influence food consumption patterns and eating habits. Cultural influences also shape food preferences; some communities may prioritize healthy eating, while others may favor fast food.

This study is the first in Saudi Arabia to examine food security, food literacy, and food intake among low-income adults, which is its primary strength. However, several limitations should be considered. Its cross-sectional design precludes the establishment of causality, making it unclear whether poor food literacy leads to food insecurity or vice versa. Participants were recruited from charitable organizations in Jeddah, which may not represent all low-income individuals in Saudi Arabia, particularly those in rural areas and other cities, or families not receiving support. Additionally, the majority of participants were women and unemployed, which could affect the generalizability of the results. Individuals receiving assistance from charities may differ from other low-income individuals who do not utilize these services, potentially introducing sampling bias. Because this study was conducted online, those without internet access or digital devices may have been excluded, and the self-administered nature of the survey increases the risk of recall or reporting bias. Furthermore, the food frequency questionnaire used did not allow for precise measurement of dietary intake.

## 5. Conclusions

This study provides the first examination of food security, food literacy, and dietary intake among low-income adults in Saudi Arabia. While most participants, particularly females, had adequate food literacy, a substantial proportion experienced severe food insecurity, especially among females, younger adults, and those engaging in less frequent meal preparation. The absence of a significant association between food literacy and food insecurity suggests that knowledge alone may be insufficient to overcome fundamental and cultural barriers to food access within this population. Socio-demographic factors, including age, education level, marital status, number of children, and income, were strongly linked to food security status, underscoring the need for interventions that address both economic and educational factors. The finding that food-insecure individuals consumed significantly fewer nutrient-rich foods further highlights their nutritional vulnerability.

These results emphasize that improving food literacy alone is not enough; rather, it is crucial to integrate food literacy programs with strategies that enhance economic stability, improve food access, and address cultural influences on food practices. Policymakers and charitable organizations should develop interventions to reduce food insecurity and promote healthy dietary behaviors in low-income communities across Saudi Arabia. Future research should adopt longitudinal designs to clarify causal relationships and further explore the connections between cultural factors, gender roles, and food-related behaviors.

## Figures and Tables

**Figure 1 foods-14-03078-f001:**
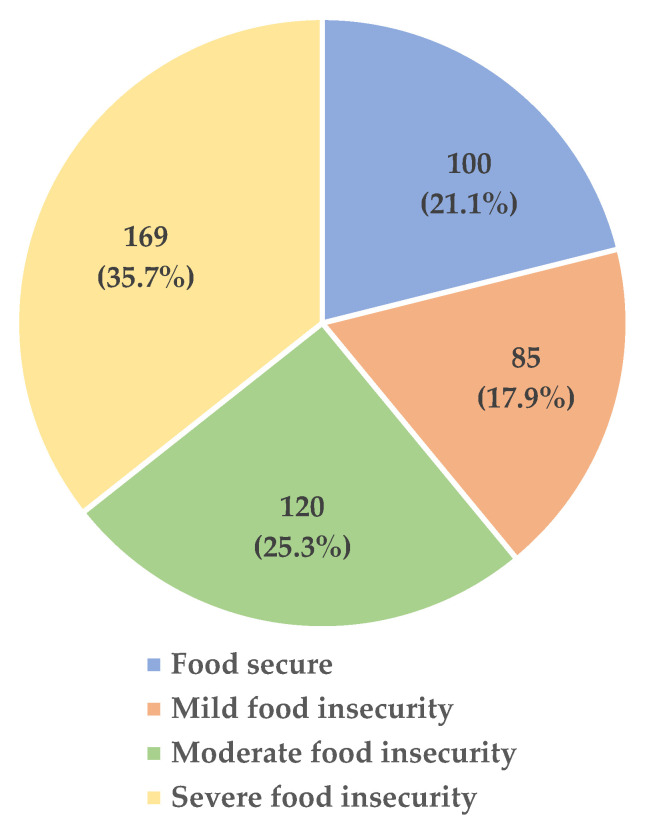
The prevalence of food insecurity.

**Figure 2 foods-14-03078-f002:**
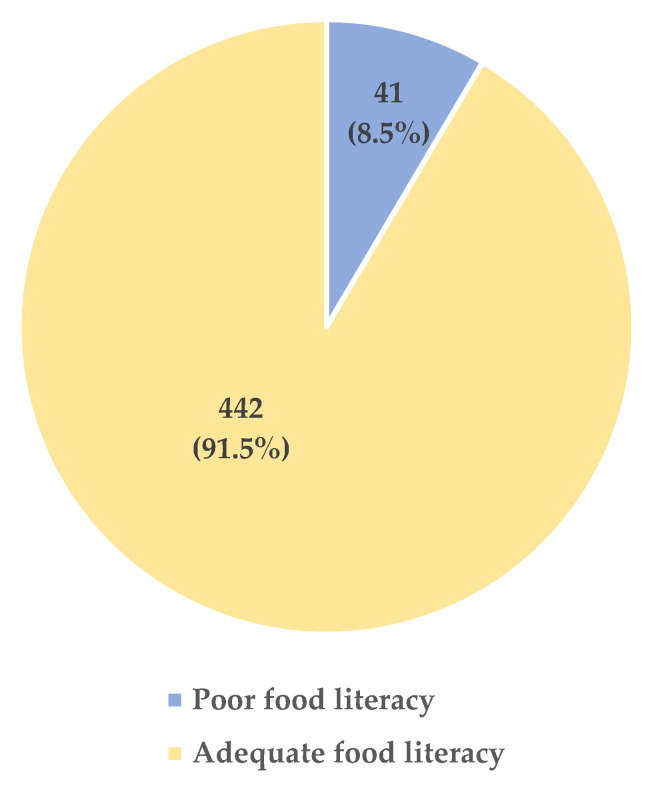
The prevalence of food literacy.

**Table 1 foods-14-03078-t001:** Demographic characteristics of the participants.

	*n*	*N* %
Gender		
Female	431	89.2%
Male	52	10.8%
Age (years)		
18–24	19	3.9%
25–34	59	12.2%
35–44	178	36.9%
45–54	166	34.4%
Above 55	61	12.6%
Level of education		
Primary school	145	30.0%
Secondary school	97	20.1%
High school	159	32.9%
Diploma	19	3.9%
Bachelor	58	12.0%
Postgraduate education	5	1.0%
Marital status		
Married	123	25.5%
Single	31	6.4%
Widowed	233	48.2%
Divorced	96	19.9%
Employment status		
Full-time employee	35	7.2%
Unemployed	428	88.6%
Part-time employee	18	3.7%
Self-employed	2	0.4%
Number of children		
No children	51	10.6%
One child	34	7.0%
2–3 Children	163	33.7%
4–5 Children	128	26.5%
>5 Children	107	22.2%
Household size		
1–2 people	32	6.6%
3–4 people	175	36.2%
Five people or more	276	57.1%
Housing structure		
Lone person	7	1.4%
Couple without children	1	0.2%
Couple with child/children	85	17.6%
Father with child/children	10	2.1%
Mother with child/children	242	50.1%
Extended family	134	27.7%
Compound family	4	0.8%
Family income, SR/month		
2000–Less than 3000	255	52.8%
3000–Less than 4000	137	28.4%
4000–Less than 5000	66	13.7%
5000 and above	25	5.2%
Living situation		
Homeowner	74	15.3%
Renter	357	73.9%
Resident of assisted living facility/residential care accommodation	52	10.8%
Meal preparation and frequency		
Prepare no meals	37	7.7%
Prepare some meals	126	26.1%
Prepare most meals	149	30.8%
Prepare all meals	171	35.4%

**Table 2 foods-14-03078-t002:** Association between food literacy and food insecurity levels.

Food Literacy	Food Insecurity				Chi-Square
	Food Secure	Mild Food Insecurity	Moderate Food Insecurity	Severe Food Insecurity	*p*-Value
Poor	6 (15%)	10 (25%)	8 (20%)	16 (40%)	0.586
Adequate	94 (21.7%)	75 (17.3%)	112 (25.8%)	153 (35.3%)	

Data are *n* (n%). Data were analyzed using the chi-square test with linear-by-linear association.

**Table 3 foods-14-03078-t003:** Comparison of food consumption between individuals with poor and adequate food literacy.

	Food Literacy		
	Poor (*n* = 41)	Adequate (*n* = 442)	*p*-Value
Fruits	0.17 ± 0.29	0.17 ± 0.23	0.141
Vegetables	0.38 ± 0.33	0.35 ± 0.32	0.555
Dairy products	0.32 ± 0.32	0.36 ± 0.34	0.612
Meat	0.33 ± 0.32	0.27 ± 0.32	0.324
Legumes	0.34 ± 0.38	0.25 ± 0.26	0.578
Bread and cereals	0.64 ± 0.39	0.52 ± 0.36	0.114
Sugar-sweetened beverages and soft drinks	0.23 ± 0.34	0.27 ± 0.32	0.141
Potato crisps or salty snacks	0.21 ± 0.3	0.25 ± 0.28	**0.04**
Sweets	0.19 ± 0.3	0.22 ± 0.27	0.068
Fast food	0.08 ± 0.09	0.15 ± 0.21	0.141

Data are presented as mean ± SD of daily consumption frequencies, calculated from the reported weekly intake of food groups and discretionary items (e.g., sugar-sweetened beverages, salty snacks, sweets, fast foods). Weekly frequencies (“I don’t eat or drink it,” “<1/week,” “1/week,” “2–3/week,” “4–6/week,” “daily”) were converted into daily equivalents by dividing by seven. Food literacy was assessed using the Short Food Literacy Questionnaire (SFLQ; range 7–52), with higher scores indicating higher levels of food literacy. “Poor” and “adequate” categories were based on the median split of the SFLQ scores. *P*-values were obtained using the Mann–Whitney test. Significant *p*-values are shown in bold.

**Table 4 foods-14-03078-t004:** Comparison of food consumption between different levels of food insecurity.

	Food Insecurity				
	Food Secure (*n* = 100)	Mild Food Insecurity (*n* = 85)	Moderate Food Insecurity (*n* = 120)	Severe Food Insecurity (*n* = 169)	*p*-Value
Fruits	0.26 ± 0.28 ^a^	0.23 ± 0.28 ^a^	0.14 ± 0.2 ^b^	0.11 ± 0.18 ^b^	**<0.001**
Vegetables	0.46 ± 0.31 ^a^	0.42 ± 0.33 ^a^	0.33 ± 0.32 ^b^	0.27 ± 0.31 ^c^	**<0.001**
Dairy products	0.49 ± 0.34 ^a^	0.44 ± 0.33 ^a^	0.33 ± 0.34 ^b^	0.24 ± 0.3 ^c^	**<0.001**
Meat	0.45 ± 0.35 ^a^	0.37 ± 0.32 ^a^	0.22 ± 0.28 ^b^	0.18 ± 0.26 ^b^	**<0.001**
Legumes	0.31 ± 0.28 ^a^	0.26 ± 0.26 ^a,b^	0.23 ± 0.25 ^b^	0.24 ± 0.28 ^b^	**0.014**
Bread and cereals	0.66 ± 0.35 ^a^	0.65 ± 0.34 ^a^	0.53 ± 0.35 ^b^	0.4 ± 0.35 ^c^	**<0.001**
Sugar-sweetened beverages and soft drinks	0.39 ± 0.37 ^a^	0.31 ± 0.32 ^a,b^	0.23 ± 0.26 ^b^	0.2 ± 0.3 ^c^	**<0.001**
Potato crisps or salty snacks	0.32 ± 0.31 ^a^	0.3 ± 0.3 ^a,b^	0.23 ± 0.25 ^b^	0.19 ± 0.27 ^c^	**<0.001**
Sweets	0.31 ± 0.31 ^a^	0.29 ± 0.31 ^a^	0.15 ± 0.21 ^b^	0.17 ± 0.26 ^b^	**<0.001**
Fast food	0.2 ± 0.2 ^a^	0.18 ± 0.23 ^b^	0.14 ± 0.22 ^b,c^	0.11 ± 0.19 ^c^	**<0.001**

Data are presented as mean ± SD of daily consumption frequencies, derived from reported weekly intake of food groups and discretionary items (sugar-sweetened beverages, salty snacks, sweets, fast foods). Weekly responses were converted into daily equivalents by dividing by seven. Food insecurity was measured with the Food Insecurity Experience Scale (FIES; score range 0–8), categorized as food secure (0), mild (1–3), moderate (4–6), or severe (7–8). *P*-values were obtained using the Kruskal–Wallis test. Significant *p*-values are shown in bold. Superscripts (a, b, c) indicate significant differences based on pairwise comparisons with Bonferroni correction. Values sharing the same superscript are not significantly different (*p* ≥ 0.05). Values with different superscripts differ significantly (*p* < 0.05).

## Data Availability

The authors declare that the data are available upon request.

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
