# Peer review of "Examining the Association Between Food Insecurity, Food Literacy, and Food Intake Among Low-Income Adults in Jeddah, Saudi Arabia: A Cross-Sectional Study"

_foods, 2025, doi:10.3390/foods14173078_

Round 1

Reviewer 1 Report

Comments and Suggestions for Authors

This article focuses on the relationship between food insecurity, food literacy and food intake among low-income adults in Saudi Arabia. The topic is closely related to the core issues of public health and nutrition, and it is the first study of its kind in this population in Saudi Arabia, demonstrating certain regional innovation. However, there are still many shortcomings:
First, the sample representativeness is seriously insufficient. The study was only conducted in Jeddah and did not cover other regions of Saudi Arabia (such as rural areas and other cities), thus failing to represent the characteristics of the national low-income group. There is a demographic imbalance, with 89.2% being female and only 10.8% male, indicating a severe gender imbalance; and 88.6% were unemployed, resulting in a single sample structure that may have missed low-income groups with income sources (such as part-time workers), limiting the generalizability of the conclusions. There may also be sampling bias, as the authors only recruited respondents through charitable organizations, which may have overly concentrated on those dependent on charitable support and ignored low-income individuals who did not receive such assistance, further reducing the representativeness of the sample.
Second, the data collection method relied solely on online questionnaires. Low-income groups may have been excluded due to the digital divide (such as lack of internet access and electronic devices), leading to the "participating respondents" not being typical representatives of the target group; and self-reporting methods are prone to recall bias (such as subjective estimation of food intake frequency).
Third, the assessment of food intake may be inaccurate. The authors only used a "frequency questionnaire" for measurement and did not adopt more precise tools (such as 24-hour dietary recall or food diaries), making it impossible to quantify the intake (such as in grams) and difficult to accurately reflect the quality of the diet.
Fourth, the explanation of the core conclusion is weak. The authors attributed the key finding of "no association between food literacy and food insecurity" only to "sample and method differences", without analyzing it in the context of Saudi culture (such as women having high food literacy but being unable to improve food security due to economic dependence; the influence of traditional dietary culture on purchasing decisions), resulting in insufficient theoretical support.
Fifth, the details of the statistical analysis are missing. The scoring method for "food intake frequency" was not clearly explained (such as how "daily" and "weekly" were converted into numerical values), and some tables had format confusion (such as the appearance of the formula symbol "Δθ" in the table in Section 3.6), affecting the readability and credibility of the results.
Sixth, the response to the study's limitations is insufficient. Although the original text mentioned "small sample size and self-reporting bias", it did not explain in the discussion the specific impact of these limitations on the conclusions (such as whether the gender imbalance overestimated the conclusion that "women have high food literacy"), nor did it propose targeted improvement directions.
It is recommended that this article be majorly revised.

Reviewer 2 Report

Comments and Suggestions for Authors

This manuscript presents a cross-sectional study assessing the relationship between food insecurity, food literacy, and food intake among low-income adults in Saudi Arabia. The topic is relevant and timely, especially given the increasing interest in food-related inequalities within vulnerable populations. However, while the study is well-intentioned and structured, the manuscript in its current form lacks the scientific depth, international contextualization, and analytical rigor expected for publication in Foods. Several methodological, analytical, and interpretative limitations reduce the strength and generalizability of the findings. Major revisions are required to improve the manuscript’s clarity, depth, and overall contribution to the literature.

Abstract

  • Line 12–25 (Abstract): Too much result detail, but weak on interpretation and contribution. Missing clarity on why this study matters!

Introduction (L28–L63)

  • Feels generic. Reads like a mix of textbook definitions and region-specific statistics.
  • L40: “…lack sufficient physical, socioeconomic, or physical access…”, Repetition of “physical.”
  • L47–54: Food literacy definition is lifted almost directly from [11]. Needs paraphrasing or better integration.

No strong research gap definition. “Insufficient studies in Saudi Arabia” is not enough.

No global framing, too Saudi-centric with no comparative lens.

References are weak: mostly local or outdated.

No justification for choice of Jeddah, why not other cities or regions?

Methods (L65–L113)

  • L66–69: Sample size explanation is too simplified. No confidence intervals mentioned.
  • Ethical Approval (L73): “XXX, XXX”  not anonymized properly.
  • L76: Data collection via online questionnaire in a low-income population? Needs justification. Many in this demographic may have limited digital access.
  • L77–78: Face validity done with only ten people which is not adequate for validation.
  • L92–98: Food literacy tool: modified SFLQ, but how was it adapted? Any pilot test for psychometric reliability?
  • L100–104: Food intake tools are vague. No details on serving size, portion standardization, or translation.

Results (L114–206)

  • The entire section is table-heavy and flat. It becomes tiring to read.
  • L121: 88.6% unemployed is extremely skewed sample. The paper should discuss this bias.
  • L138–150 (Table 2): Many comparisons with no correction for multiple testing! this inflates Type I error risk.
  • L169–177 (Table 4): No significant association found, yet the conclusion draws weight from this which feels contradictory.

Tables:

  • Too many long tables back to back. Consider grouping or splitting to improve readability.
  • No visualization: a simple bar chart or trend line would help communicate results more intuitively.

Discussion (L207–305)

  • L208–220: Repeats the results without insight.
  • L248–258: Mention contradictions and similarities without explaining or justifying the reason! No resolution or deeper analysis.

This section suffers from “study dumping” listing other findings without integration.

Almost no mechanisms or theoretical interpretation. Why does food literacy not correlate with intake here? Is there cultural moderation? Resource limitation?

No international comparisons (e.g., data from Egypt, Iran, Brazil) — this isolates the findings.

Conclusion (L307–317)

  • Generic and unimpactful.
  • L314–316: Says more research is needed, but gives no specific direction. What kind? longitudinal? interventions? mixed methods?

References

  • Total 28 refs. That’s very low for a full-length paper.
  • Too many self-citations or local studies.
  • Lacks foundational or theoretical works on food literacy and food security (e.g., works by Vidgen, Thomas, or Nutbeam).
  • No references from South Asia, Africa, or Latin America! This makes the context weak globally.

Reviewer 3 Report

Comments and Suggestions for Authors

The manuscript entitled ''Examining the Association Between Food Insecurity, Food Literacy, and Food Intake Among Saudi Low-income Adults'' reveals a significant negative correlation between food insecurity and dietary intake among Saudi Arabia's low-income population (35% severe food insecurity rate) through a cross-sectional study, but fails to confirm the moderating effect of food literacy (P=0.586). While the research design is fundamentally scientific, it suffers from sampling bias (89.2% female participants), methodological inconsistencies, insufficient depth in discussion, and logical inconsistencies in conclusions, necessitating major revision. 

Comments to the Author:

Scientific Rigor:

Question 1:Severe gender imbalance (89.2% female) in sampling (Table 1) undermines the generalizability of findings on food literacy and food insecurity. Statistical adjustments or stratification analysis must be added.

Question 2:"Adequate food literacy" (91.5%, Figure 2) contradicts Saudi national data (Bookari 2023 cited). Sampling bias from charity-dependent recruitment requires clarification.

Question 3:It is necessary to clarify the definition of low-income groups, and the official low-income standards of Saudi Arabia can be referred to.

Question 4:This study is a single-time-point cross-sectional design, which cannot infer causality. It should be clearly stated in the title, abstract, limitations.

Introduction

Question 5:"Food literacy" is not operationally defined in the Saudi context.

Question 6:Background omits key drivers of food insecurity in Saudi Arabia; Cite national economic surveys, for example, GASTAT reports.

Question 7:The hypothesis that “food literacy may buffer food insecurity” is weakly framed; provide a systematic overview of current evidence and gaps.

Method

Question 8:Self-reported online questionnaires (Section 2.2.1) introduce recall and social desirability biases, especially for food intake frequency (e.g., "never" to "daily" scales in Section 2.2.3). No mitigation strategies are mentioned. Add validation steps or discuss limitations in bias impact.

Question 9:The methods claim non-parametric analysis (Mann-Whitney U and Kruskal-Wallis tests sin Section 2.3), but Table 6 uses parametric superscripts (a,b,c) to denote significant differences, indicating confusion in test selection. Correct the analysis to purely non-parametric notation or justify parametric assumptions.

Novelty

Question 10:Although claimed as “first in Saudi Arabia”, no table contrasts international findings; add a novelty table.

Question 11:As the first Saudi study on this topic, emphasize unique socio-religious factors (e.g., Ramadan fasting, zakat systems) affecting food behaviors.

Discussion

Question 12:The finding that “severely food-insecure consumed fewer SSB/fast food” contradicts Begley et al.; provide deeper discussion of discrepancies.

Question 13:The paradox that higher-literacy participants consumed slightly more crisps is under-discussed; propose explanations (e.g., affordability).

Question 14:Critical data on 18-24 yr olds - highest severe insecurity (52.6%, Table 3) and poorest literacy (31.6%, Table 2) - is merely descriptive (Lines 311-313). Contrast this with global evidence and hypothesize drivers.

Conclusions

Question 15:The conclusion calls for “larger samples” without power-based justification; provide post-hoc power or desired sample calculation (Estimate how many subjects are needed to safely detect real differences or associations).

Question 16:The suggestion for longitudinal follow-up lacks specific research questions and design; state testable hypotheses.

Question 17:The conclusion offers no actionable recommendations for charities or policymakers; add intervention strategies grounded in findings.

Supplementary Opinions

Question 18:The manuscript features an insufficient variety of Figures and an excessive number of Tables. It is recommended to diversify the types of Figures used, and some Tables could be relocated to the supplementary materials.

Question 19: Standardize all references per journal guidelines.

Reviewer 4 Report

Comments and Suggestions for Authors

Food insecurity is a deeply relevant topic in scientific debates and is a significant social concern. Equally important is the relationship between food insecurity and food literacy.

Therefore, this is a manuscript that focuses on a clearly current and pertinent topic.

From a methodological standpoint, I have no criticisms of the work.

However, it seems to me that the manuscript presents a clear analytical weakness in the discussion of the results and even more so in the conclusions. Therefore, a significant improvement in the reflective dimensions is essential, particularly in these parts of the work.

On the other hand, the study fails to consider the cultural dimensions that involve "low-income Saudi families." It seems to me that knowledge of these dimensions is essential to understand precisely the "prevalence of food insecurity and food literacy and examine food intake among low-income Saudi families."

Reviewer 5 Report

Comments and Suggestions for Authors

  1. 95% level of significance?.. I think authors mean confidence. 
  2. Which online calculator?
  3. Was the name of Biomedical Ethics Committee purposely left out?
  4. Were participants compensated or required to complete the survey?
  5. Who completed the surveys? Was it the person who would typically prepare meals? 
  6. Are the food frequency questionnaires really measuring intake or are they assessing intake?
  7. How many participants were invited to take the survey? / What is the percentage completion rate?
  8. I like the food insecurity graph, but might be helpful to have different colors for the categories vs shades of blue; different colors would make easier to interpret.
  9.  Section 3.5 describes statistical methods - this should go in methods, not results.
  10. Table 5 - how were these components selected or calculated?
  11. Line 203: "These results indicate that higher levels of food insecurity are 203
    associated with reduced intake of a variety of food items." Be careful using 'higher' vs 'more severe' 
  12. Given this is cross sectional data, the authors cannot confirm directionality. In the discussion it is possible food insecurity and literacy impacted whether individuals prepared meals. The authors state "Similarly, individuals who did not prepare meals indicated high rates of poor food literacy and severe food insecurity." - the authors should discuss the potential for the opposite direction as well to be true.
  13. "Similarly, individuals who did not prepare meals indicated high rates of poor food literacy and severe food insecurity." is misleading given the following sentence says there is no association between food literacy and food security.
  14. "Additionally, higher levels of food insecurity were associated with a reduced intake of a variety of food items, including fruits, vegetables, dairy products, and meat." Again, suggest using 'more severe' through out paper rather than 'higher' in regards to food insecurity.
  15. "A similar intervention aimed at improving food literacy among low-income adults reported similar participant characteristics, with most participants being female and aged between 36- 45, although their educational levels differed, as a greater proportion held diploma degrees" - is there an intervention in the present study?? Remove 'similar' if not.
  16. Improve sentence: "No significant associations were found in the current findings between food literacy and education level or family income..."
  17. Authors need to be careful interpreting the results from food frequency questionnaires - they cannot be used to quantify the amount individuals eat; can only tell how many times an item was consumed. This could certainly contribute to the discrepancies in results from this study compared to others (if others used methods that could quantify items such as 24 hour recalls, etc.)
  18. Authors need to better explain why results are different from other studies - something about this population? something about the methods? Be more specific.
  19. "This divergence in the findings may be attributed to this study’s focus on low-income families." Can authors say more about this in relation to why individuals with adequate food literacy ate more salty food?
  20. For limitations, is this a small study? It has hundreds of people and exceeded the power calculation. Also, using FFQ is a limitation vs self-response. FFQs cannot provide the quantity in which an individual ate, it can only provide the amount of times the food was consumed. This is a major limitation to interpreting the results from the study.

Round 2

Reviewer 2 Report

Comments and Suggestions for Authors

Dear Authors,

Thank you for revising the manuscript. While many of my previous concerns have been addressed, several points remain only partially addressed or unresolved:

  1. Abstract: Interpretation and contribution improved, but novelty and significance could still be stated more explicitly.
  2. Introduction: Global framing remains limited; references from South Asia, Africa, or Latin America are still minimal.
  3. Sample Size Calculation: Formula description remains brief! please include the exact formula and parameters used.
  4. SFLQ Adaptation: Adaptation process is minimally described, no pilot testing was conducted, this limitation should be acknowledged.
  5. Food Intake Tools: Portion standardization and translation procedures remain unclear.
  6. International Context: Only one new comparison (Canada) added, more diverse contexts would strengthen discussion.

Author Response

Thank you for your detailed and insightful comments, which have been invaluable in improving the quality and clarity of our manuscript. We appreciate the opportunity to revise our work and have addressed each of your comments carefully. Below, we provide a point-by-point response, along with the corresponding revisions made in the manuscript.

Comment 1: Abstract: Interpretation and contribution improved, but novelty and significance could still be stated more explicitly.

Response: Thank you for this valuable suggestion. We carefully considered this point; however, given the strict word limit for the abstract, we have modified the sentence, emphasizing the novelty of our study and its significance in contributing to the literature on food literacy and food insecurity, page 1, lines 12–15 and 1, lines 26-28.

Comment 2: Introduction: Global framing remains limited; references from South Asia, Africa, or Latin America are still minimal.

Response: Additional references from studies conducted in South Africa and Latin America have now been included to strengthen the global framing of the introduction, page 2, lines 45–48.

Comment 3: Sample Size Calculation: Formula description remains brief! please include the exact formula and parameters used.

Response: We appreciate this observation. The exact formula and parameters used for the sample size calculation have been added to the Methods section (page 3, lines 90–92).

Comment 4: SFLQ Adaptation: Adaptation process is minimally described, no pilot testing was conducted, this limitation should be acknowledged.

Response: Thank you for this comment. We would like to clarify that the Short Food Literacy Questionnaire (SFLQ) used in our study was adapted from the version previously pilot tested by Bookari et al. In their study, the tool was subjected to face and content validity by a panel of experts and underwent pilot testing with a sample of parents, leading to refinement of several items for cultural appropriateness. In our study, we culturally adapted the questionnaire based on the Saudi context. However, an expert panel evaluated face and content validity to ensure accuracy and relevance. This has been clarified in the Methods section (page 3, lines 102–104) and (page 3, lines 120–121).

Comment 5: Food Intake Tools: Portion standardization and translation procedures remain unclear.

Response: Further clarification has been added to the method section regarding the portion size standardization and the translation  (page 4, lines 130–133).

Comment 6: International Context: Only one new comparison (Canada) added, more diverse contexts would strengthen discussion.

Response: We have now included comparisons with additional studies from diverse international contexts, including Iran, Australia, and western Honduras to enrich the discussion and provide broader cultural perspectives (page 11, lines 256–258), (page 11, lines 265–267), (page 12, lines 290–294),

Reviewer 3 Report

Comments and Suggestions for Authors

Based on the revised manuscript and your responses, I confirm that the authors have adequately addressed all concerns raised during the review process. The manuscript now meets the journal’s scientific standards and requires only final grammar and language polishing before publication. Recommend acceptance after thorough proofreading to ensure clarity and consistency throughout the text.

Author Response

Comment 1: Based on the revised manuscript and your responses, I confirm that the authors have adequately addressed all concerns raised during the review process. The manuscript now meets the journal’s scientific standards and requires only final grammar and language polishing before publication. Recommend acceptance after thorough proofreading to ensure clarity and consistency throughout the text.

Response: We sincerely thank the reviewer for the positive feedback. The manuscript has been professionally proofread, and a proofreading certificate has been submitted with the revised version. We also carefully checked the grammar and consistency throughout the manuscript to ensure clarity.

Reviewer 4 Report

Comments and Suggestions for Authors

The new version of the manuscript demonstrates an effort to incorporate the suggestions I presented in my first review. It seems to me that the work has gained argumentative polish, specifically in the discussion of the results and conclusions. I'm certain that if the authors had expanded on the cultural dimensions a bit more, the manuscript would have gained a framework that could allow for a better understanding of some aspects of the argumentative line.

Author Response

Comment 1: The new version of the manuscript demonstrates an effort to incorporate the suggestions I presented in my first review. It seems to me that the work has gained argumentative polish, specifically in the discussion of the results and conclusions. I'm certain that if the authors had expanded on the cultural dimensions a bit more, the manuscript would have gained a framework that could allow for a better understanding of some aspects of the argumentative line.

Response: Thank you for your feedback. We have incorporated additional discussion points to reflect how cultural context may influence food literacy and food insecurity (page 2, lines 62–70) and (page 13, lines 317–321)

Reviewer 5 Report

Comments and Suggestions for Authors

Reviewer comments and questions appear to be addressed. Nice work.

Author Response

Comment: Reviewer comments and questions appear to be addressed. Nice work.

Response: We thank the reviewer for this encouraging feedback and are pleased that the revisions have addressed the concerns raised.